# Prevalence, Antibiotic-Resistance, and Replicon-Typing of *Salmonella* Strains among Serovars Mainly Isolated from Food Chain in Marche Region, Italy

**DOI:** 10.3390/antibiotics11060725

**Published:** 2022-05-28

**Authors:** Ilaria Russo, Daniela Bencardino, Maira Napoleoni, Francesca Andreoni, Giuditta Fiorella Schiavano, Giulia Baldelli, Giorgio Brandi, Giulia Amagliani

**Affiliations:** 1Department of Biomolecular Sciences, University of Urbino Carlo Bo, 61029 Urbino, Italy; i.russo6@campus.uniurb.it (I.R.); daniela.bencardino@uniurb.it (D.B.); francesca.andreoni@uniurb.it (F.A.); giulia.baldelli@uniurb.it (G.B.); giorgio.brandi@uniurb.it (G.B.); 2Istituto Zooprofilattico Sperimentale dell’Umbria e Delle Marche “Togo Rosati”, 06126 Perugia, Italy; m.napoleoni@izsum.it; 3Department of Humanities, University of Urbino Carlo Bo, 61029 Urbino, Italy; giuditta.schiavano@uniurb.it

**Keywords:** *Salmonella* serovars, antibiotic resistance, *tet* genes, multidrug-resistant (MDR), extensively drug-resistant (XDR), antimicrobial resistance genes (ARGs), plasmid profile, PCR-based replicon typing (PBRT), food chain

## Abstract

Nontyphoidal salmonellosis (NTS) is the second most commonly reported gastrointestinal infection in humans and an important cause of food-borne outbreaks in Europe. The use of antimicrobial agents for animals, plants, and food production contributes to the development of antibiotic-resistant *Salmonella* strains that are transmissible to humans through food. The aim of this study was to investigate the presence and the potential dissemination of multidrug-resistant (MDR) *Salmonella* strains isolated in the Marche Region (Central Italy) via the food chain. Strains were isolated from different sources: food, human, food animal/livestock, and the food-processing environment. Among them, we selected MDR strains to perform their further characterization in terms of resistance to tetracycline agent, carriage of *tet* genes, and plasmid profiles. Tetracycline resistance genes were detected by PCR and plasmid replicons by PCR-based replicon typing (PBRT). A total of 102 MDR *Salmonella* strains were selected among the most prevalent serovars: *S*. Infantis (n = 36/102), *S*. Derby (n = 20/102), *S*. Typhimurium (n = 18/102), and a monophasic variant of *S*. Typhimurium (MVST, n = 28/102). Resistance to sulfisoxazole (86%) and tetracycline (81%) were the most common, followed by ampicillin (76%). FIIS was the most predominant replicon (17%), followed by FII (11%) and FIB (11%) belonging to the IncF incompatibility group. Concerning the characterization of *tet* genes, *tet*B was the most frequently detected (27/89), followed by *tet*A (10/89), *tet*G (5/89), and *tet*M (1/89). This study showed the potential risk associated with the MDR *Salmonella* strains circulating along the food chain. Hence, epidemiological surveillance supported by molecular typing could be a very useful tool to prevent transmission of resistant *Salmonella* from food to humans, in line with the One Health approach.

## 1. Introduction

Nontyphoidal salmonellosis (NTS) is a common infection mainly caused by the ingestion of food or beverages contaminated by several zoonotic serovars with the potential to interact with human and animal hosts [1,2]. The most prevalent serovars responsible for human illnesses acquired in the European Union during 2019 were, in decreasing order, *S*. Infantis, *S*. Enteritidis, the monophasic variant of *S*. Typhimurium (MVST), *S*. Typhimurium, and *S*. Derby [3]. 

Food-producing livestock and domestic pets are most often sources of NTS food-borne outbreaks [1]. Indeed, NTS is linked to the consumption of *Salmonella*-contaminated food mostly from poultry, pork, and egg products. However, in the last few decades, the sporadic occurrence of these microorganisms was also detected in fruit and vegetable produce [3,4]. Furthermore, poor hand washing and contact with infected pets are some of the contamination routes. Thus, as a consequence of the growth in consumption of products of animal origin and the integration of companion animals in households, there is an increased potential for exposure to *Salmonella* via the food chain [4]. 

When contaminated food is ingested, the pathogen attacks and invades the intestinal epithelium of the distal ileum triggering sickness that appears as acute gastroenteritis within 4 to 72 h. Fever, chills, nausea, vomiting, abdominal cramping, headache, and diarrhea are the main symptoms [1,4]. Usually, in healthy individuals, NTS is a self-limited disease resolving in a few days without medical intervention, but some patients may develop chronic sequelae such as reactive arthritis or irritable bowel syndrome [4,5]. On the other hand, in vulnerable patients (immunocompromised, very young or elderly persons) NTS infection can systematically spread to other body organs causing febrile illness [2,3,4]. Hence, NTS is a public health concern representing an economic burden for both developed and developing countries, due to costs associated with surveillance, prevention, and treatment of disease [1,6,7].

The rise in the occurrence of antimicrobial-resistant (AMR) strains in the food chain increases the risk associated with this zoonotic infectious disease minimizing treatment options and increasing human mortality [8,9]. The emergence of antimicrobial resistance in the food chain is considered a cross-sectoral problem due to: (i) the misuse and overuse of antibiotics in agricultural and livestock production, (ii) the dissemination of antimicrobial resistance genes (ARGs) among bacteria intentionally added during processing (e.g., starter cultures and probiotics), (iii) the post-contamination by the environment, after food processing, (iv) the cross-contamination with AMR bacteria colonizing other foods during industrial processing and handling by the consumer [10,11]. 

The selective effects of antimicrobial use accelerate the natural development of resistance mechanisms activated by bacteria, and the acquisition of ARGs can occur at any stage of the food chain facilitated by mobile genetic structures such as plasmids, integrons, and transposons, in addition to vertical transfer [11,12]. Indeed, the presence of resistance determinants in foodstuffs increases the gene pool by which pathogens can achieve and transfer ARGs to other bacteria, thus representing an indirect risk to public health [11]. 

Horizontal gene transfer has been repeatedly described in *Salmonella*, especially for tetracycline resistance genes (*tet*) because some of the most frequently detected, such as *tet*A and *tet*G, are located on mobile genetic elements [13]. As observed in the latest report of the European Food Safety Authority (EFSA), Italy showed a high level of resistance toward tetracyclines, confirming the alarming effects caused by the abuse of these broad-spectrum agents in both human and veterinary medicine [3,13]. In light of recent scientific issues on antimicrobial resistance, the EFSA updated technical specifications for the implementation of molecular typing methods during routine monitoring. EFSA proposal aims to reach harmonised surveillance of antimicrobial resistance in food-producing animals and derived food to ensure continuity in following up the dynamic evolution of AMR foodborne pathogens [14]. Mobile genetic elements play a pivotal role in the dissemination of antimicrobial resistance along the food chain and their study is crucial to better understand their epidemiology, apply control measures and reduce the presence of the pathogen in food [11].

In this study, we selected a group of strains isolated in the Marche Region including those serotypes mainly detected in *Salmonella* strains. Within this collection, we evaluated their distribution in each one of the niches considered (animal, food, environment, and human), their antimicrobial resistance, and plasmid profiles in order to obtain a general view of isolates circulating in our region. Our aim was to investigate the presence and the potential dissemination of AMR *Salmonella* isolates via the food chain.

## 2. Results

### 2.1. Salmonella Strains

A total of 102 AMR Salmonella strains, of serovars S. Derby (n. 20), S. Typhimurium (n. 18), MVST (n. 28), and S. Infantis (n. 36) were collected and analysed for this study. Strains of animal origin (n. 16) were from pigs (n. 5) and poultry (broiler, n. 8; pigeon, turkey, and laying hen, n. 1 each); food samples (n. 35) included meat products (pork meat, n. 20; bovine, n. 4; chicken, n. 7), and mollusks (n. 4). Environmental samples (n. 9) were swabs from food processing rooms (n. 2), slaughter rooms (n. 4), and poultry farms (n. 3) (Table 1). Human clinical *Salmonella* strains (n. 42) were isolated from faeces (n. 37), blood (n. 2) urine (n. 2), and other clinical samples (n. 1) (Table 1).

### 2.2. Antibiotic Susceptibility Profiles

All *Salmonella* strains analysed in this study were resistant to one or more antibiotic classes, except for one unique environmental *Salmonella* isolate, that was intermediate resistant to streptomycin and ciprofloxacin and sensitive to all other antibiotics tested. Nevertheless, this last strain was included in our collection in order to obtain a reasonable number of environmental strains, in light of the limited size of this niche. The Appendix A showed the distribution of antibiotic resistance detected among strains isolated from the different niches. Independently of isolation origin and serovar, a high rate of resistance was recorded to sulfisoxazole (88 out of 102 strains, 86%) and tetracycline (83/102, 81%), followed by ampicillin (78/102, 76%), streptomycin (61/102, 60%) and nalidixic acid (38/102, 37%). Finally, 26 strains (25%) were resistant to cefotaxime and confirmed as extended-spectrum β-lactamase (ESBL)-producing *Salmonella*. Moreover, a high proportion of *Salmonella* isolates (42%; 43/102) showed intermediate sensitivity to ciprofloxacin, a 2nd generation fluoroquinolone, whereas meropenem resistant strains were not detected (Figure 1A).

As shown in Figure 1B, the majority of strains (47/102, 46%) were resistant to three or more antibiotic classes, hence they were classified as multidrug-resistant (MDR) [15], whereas the 44% of strains (45/102) were classified extensively drug-resistant (XDR) because they were resistant to at least one agent in all but two or fewer antimicrobial categories (i.e., bacterial isolates remain susceptible to only one or two categories) [15]. The remaining 9 strains showed resistance to up two antibiotics of two categories.

The resistant phenotype was not equally distributed among serovars. In *S*. Infantis, strains resistant to tetracycline, sulfamides, streptomycin, nalidixic acid, pefloxacin, ampicillin, and cefotaxime were the most frequently observed, while reduced sensitivity to ciprofloxacin was noticed among a relevant number of strains which were resistant to other antibiotics. Streptomycin, sulfisoxazole, and tetracycline were the most observed resistance phenotypes in *S*. Derby. *S*. Typhimurium and MVST showed similar antibiotic resistance profiles: ampicillin, sulfisoxazole, streptomycin, and tetracycline were the most diffused resistance phenotypes in both serovars, and *S*. Typhimurium also showed a certain proportion of strains resistant to chloramphenicol (Figure 2).

### 2.3. Tetracycline Resistance Genes

Among the 83 resistant strains and the three strains with intermediate sensitivity to tetracycline, a *tet* gene was determined in 43 strains, while the remaining 43 strains tested negative for each target. The gene most frequently detected was *tet*B (27/86), followed by *tet*A (10/86) and *tet*G (5/86), while *tet*M was found in a single strain isolated from humans and was in combination with *tet*B. On the other hand, *tet*C and *tet*D were not detected (Table 2). 

To note, almost all determinants were heterogeneously distributed both in terms of serovars and niches. However, *tet*A was not found in MVST that was instead characterised by the carriage of *tet*B for the 97% of tetracycline-resistant strains. Further, *tet*G gene was detected only in *S*. Typhimurium strains, and all were isolated from food (Table 2).

### 2.4. Replicon Typing

Among the 102 strains analysed by PBRT only 12 out 30 replicons were detected (HI2, I1α, N, I2, FIB, I1γ, A/C, FIIS, X1, FIB KN, FII, X4), as described in Figure 3A. 

Overall, 58 strains were untypeable because they were negative for all replicons whereas 28 strains were positive for 1 replicon, 11 for 2 replicons, and 5 for 3 replicons. FIIS was the most predominant replicon (17%), followed by FII (11%), and FIB (11%). However, we observed specific replicon distribution among strains belonging to the serovars considered in this study. Indeed, the IncX group was detected in *S*. Infantis and MVST, IncF in *S*. Typhimurium, *S*. Infantis, and MVST whereas IncI and IncH were detected only in *S*. Derby. 

Furthermore, replicons IncI and IncH were detected in strains isolated from human and food samples, IncF group was the most detected in strains isolated from animal, food, and human samples whereas, among strains isolated from the environment, only the IncX group was detected (Figure 3B). 

Results of the PBRT analysis allowed us to define 18 PBRT profiles (i.e., combinations of replicons). The prevalent profile included strains with the single replicon FIIS (n = 11/102), followed by I1α (n = 4/102), and FIB,FII (n = 4/102). All PBRT profiles detected in the present study comprised strains of the same serovar, except for HI2 detected in both *S*. Derby and MVST, and X1 detected in MVST and *S*. Infantis (Figure 3B). Multireplicon status (two or more replicons) was recorded in 16 strains, with a maximum of three replicons in five MVST strains. Multireplicon profiles were the following: FIB,A/C,FII (n = 1), I1α,FIB,FII (n = 1), and FIB,FIIS,FII (n = 3). Four out of these five strains having the multireplicon profile were resistant to tetracycline, three of which carried *tet*B gene or *tet*B combined with *tet*M.

In this study, the possible association between the Inc group and resistance phenotypes was also investigated. Results of statistical analysis (Hypothesis test–difference between two frequencies in large samples) showed significant association between IncF presence and resistance to sulfisoxazole (*p* < 0.00001), chloramphenicol (*p* < 0.0001), streptomycin (*p* < 0.01), ampicillin (*p* < 0.01) and tetracycline (*p* < 0.05).

## 3. Discussion

In this study, the characterisation of *Salmonella* strains belonging to the main serovars isolated from the food chain in the Marche Region (Central Italy) is reported. The decision to select *S*. Infantis, *S*. Derby, *S*. Typhimurium, and MVST as the target of the investigation was derived from taking into account the epidemiology of main serovars circulating at the regional level in both veterinary and human samples [16,17,18,19,20,21]. The dominance of these four serovars was consistent with that observed at the national and European levels, as described by the last reports of Entervet and EFSA/ECDC [3,22]. 

With the exception of *S*. Typhimurium which presents a global distribution [23], geographical differences emerged comparing the serovars predominant in Europe with those in the USA, with the prevalence of *S*. Enteritidis and *S*. Newport for the latter [23,24].

All serovars considered in the present work are commonly found along the food chain and they are able to cause infections in humans [23]. *S*. Typhimurium and its monophasic variant (MVST) are associated with pig and pork meat [3]. In food-processing environments, these pathogens can spread throughout processing lines and, thus, reach utensils, surfaces, and hands of employees [25]. Indeed, the prevalence of *S*. Typhimurium and its monophasic variant is similar in both farms and slaughterhouses, and strains isolated from swine and humans show important correlations in terms of molecular profiles [23,26]. *S.* Derby is continuously detected in pig, slaughter, and pork meat [3], and several studies demonstrated correlations between strains isolated from pigs and those of human origin [27,28]. *S*. Infantis is reported as the most frequent poultry-adapted *S*. enterica serovar with an increasing occurrence in broiler flocks, derived meat, and breeding hens [29,30]. 

From a health perspective, these findings are remarkable because the spread along the food chain of serovars with the ability to cause infections in humans represents a great concern. Indeed, *Salmonella* is the second pathogen responsible for food-transmitted diseases and is an important cause of foodborne outbreaks in the EU [3]. 

Over recent years, the risk associated with the increasing incidence of these serovars in humans and animals has been complicated by the spread of MDR clones in several European countries, including Italy [31,32]. The overuse of antimicrobial agents in therapy and prophylaxis of human and animal infections, as well as growth-promoting agents in food animal production, favored the emergence of *Salmonella* strains resistant to several classes of antibiotics, including those used as the primary choice for clinical treatment [33]. In light of this, we presented new and updated data concerning not only the occurrence of the most prevalent serovars detected at the regional level but also their molecular features in terms of antibiotic resistance and plasmid profiles. We found a heterogeneous distribution of phenotypic resistance among the serovars investigated, confirming also the typical multi-resistance pattern frequently determined in *Salmonella* strains circulating in Europe [34].

Tetracyclines, which are broad-spectrum drugs widely used in human and veterinary medicine, represent one of the main agents toward which *Salmonella* developed a high level of resistance [35]. As reported by the latest ESVAC (European Surveillance of Veterinary Antimicrobial Consumption) report, of the overall antimicrobials used in the 31 countries in 2020, the largest amounts were for penicillins (31.1%), tetracyclines (26.7%), and sulfonamides (9.9%) [36]. In our study the highest resistance rate was determined toward sulfisoxazole, followed by tetracycline, both classified as highly important antibiotics for human medicine [37]. Thus, the responsible use of them is strongly recommended in order to keep the associated risk as low as possible [38]. 

This study demonstrated wide dissemination of tetracycline resistance in *Salmonella* strains (80%) along with the considered food chain settings. This was in accordance with the last report released by EFSA where a rate of tetracycline resistance higher than the European average was described for Italy [34]. Among all tetracycline-resistant strains of our collection, 49% harboured one or more *tet* genes (*tet*A, *tet*B, *tet*C, *tet*D, *tet*G, *tet*M), matching with other studies carried out both in Italy and in Europe [39,40,41]. The fact that only 49% of strains resistant to tetracycline were positive for at least one *tet* gene is not surprising because this resistance could be due to different genes or other mechanisms such as mutations within the ribosomal binding site, activity of efflux pumps or enzymatic inactivation of tetracycline drugs [42]. In agreement with other authors, the genes most frequently detected in the present work were *tet*A and *tet*B belonging to Group-I and associated with an efflux pump mechanism [43,44,45,46]. On the contrary, all strains of our collection were negative for *tet*C and *tet*D. This result is not uncommon and could be due to the low ability of those genes to confer resistance to tetracycline [46], leading to their infrequent detection. The presence of these genes in strains isolated from animals increases the risk associated with circulating *Salmonella* strains in livestock for many reasons. Firstly, animals can transfer these strains to humans when farmers come into close contact with them. Secondly, animals can release strains through faecal material contributing to the spread of ARGs within livestock, and cross-contamination events, due to low hand hygiene compliance which can occur during processing.

Remarkably, the localization of many ARGs on mobile genetic elements (plasmids, integrons, and transposons) makes them easily transferred to both other bacteria and the environment [47]. This is the case of *tet*A and *tet*B which are the most frequently detected genes in our collection, confirming their increased ability to spread.

Hence, screening programmes supported by molecular typing methods are very useful for the epidemiological tracing of serovar distribution in different sources along the food chain, and also for the prompt identification of potential risks for the health of farmers, food handlers, and consumers. For this reason, all strains were further characterised by PBRT in order to understand the distribution of plasmid-related antimicrobial resistance determinants based upon replicon types.

It is known that a variety of plasmid families are frequently found in *Enterobacterales* promoting the rapid and consistent dissemination of ARGs [48]. The most common Inc replicon types found in our collection were IncFIIS, followed by IncFII and IncFIB. The dominance of IncF plasmids in AMR *Enterobacterales* isolated from humans and animals has been widely assessed [48,49]. It was also demonstrated that IncF plasmids carry multireplicon patterns where one replicon is strongly conserved while the others are free to diverge. So, the selective pressure imposed leads plasmids to duplication and dissemination [49]. Furthermore, the low copy number IncF plasmids contain the ADP-ribosylating toxin, SpvB causing the systemic virulence typical of some serovars such as *S*. Derby and *S*. Typhimurium [50]. For all of these reasons, the implementation of molecular investigation in surveillance programmes achieves more and more relevance.

Besides the described dominance of IncF, previous studies carried out in Italy recorded the occurrence of different plasmid replicons. Dionisi and colleagues [32] characterised *S*. Infantis isolated from different sources, between 2005 and 2011, identifying IncHI1 as the dominant replicon. Instead, Franco and colleagues [31] investigated *S*. Infantis isolates from different broiler chicken flocks detecting IncP replicon in all of them. Interestingly, Di Cesare and colleagues [51] reported a large variety of Inc-plasmids among isolates from animals and foodstuffs, highlighting the diversity of the *Salmonella* serovars. To note, IncHI2 and IncFIIS were prevalent and detected only in specific serovars suggesting that their distribution could be serovar-dependent [51]. This issue was found in accordance with that observed in our study where IncH and IncI were found to be associated only with *S*. Derby while IncX was detected only in MVST and *S*. Infantis strains originated from food and human samples, suggesting their potential transmission along the food chain.

A large part of strains was untypeable by PBRT, and what could be a limitation for the study actually highlights the need of monitoring circulating strains continuously, in order to follow the dynamic evolution of *Salmonella*. This foodborne pathogen, as well as other bacteria, has a tendency to modify its molecular elements to obtain advantageous adaptability resulting in the escape of current assays. 

It is clear that a better knowledge of molecular aspects in terms of resistance determinants and plasmid content can be helpful to understand how resistant bacteria spread within food settings.

## 4. Materials and Methods

### 4.1. Study Design and Selection of Strains

This study considered *Salmonella* strains isolated from different samples collected in the Marche Region, Italy, between 2015 and 2021, in the framework of the official controls provided by the Regulation (EC) No 2073/2005 on microbiological food safety criteria [52] the National Poultry monitoring plan [53,54] and during self-monitoring controls both in food [52] and veterinary sectors [54].

*Salmonella* strains were collected from microbiological analysis performed as part of these controls and serotyped at the Regional Reference Centre for Enteric Pathogens of the Istituto Zooprofilattico Sperimentale dell’Umbria e delle Marche, section of Tolentino (Italy). 

Only AMR strains of the main circulating serovars were selected including *S*. Derby (n = 20/102; 20%), *S*. Infantis (n = 36/102; 35%), *S*. Typhimurium (n = 18/102; 18%), and MVST (n = 28/102; 27%). The dominance of these serovars was derived from epidemiological information about the main serovars circulating in both veterinary and human samples [3,21]. Among them, we selected MDR strains to perform their further characterisation in terms of resistance to tetracycline agent, carriage of *tet* genes, and plasmid profiles.

Isolation sources included different points of the food chain, from farm and veterinary environments to food and food processing plants. Animal samples were obtained mainly from pigs and poultry (faeces, dust, and boot swabs collected on farms, viscera collected during necropsy) while food samples included meat and fish products. Environmental samples were obtained by collecting swabs from food processing rooms, slaughter rooms, and poultry farms. Finally, human *Salmonella* strains were isolated from people with gastrointestinal symptoms, some of them hospitalized. These strains were sent to the Regional Reference Centre for Enteric Pathogens from the Regional hospitals’ analysis laboratories participating in Enter-Net surveillance for Marche Region. Strain origin and serotypes were reported in Table 1.

### 4.2. Serotyping Analysis and Antibiotic Susceptibility Testing

All *Salmonella* isolates from either veterinary, food or human samples were serotyped according to ISO/TR 6579-3:2014 [55].

Antimicrobial susceptibility of the *Salmonella* strains was determined by the disk diffusion method, according to the Clinical and Laboratory Standards Institute guidelines (CLSI, 2021) toward the following antibiotic agents: ampicillin (AMP, 10 μg), cefotaxime (CTX, 30 μg), ceftazidime (CAZ, 30 μg), amoxicillin+clavulanic acid (AMC, 30 μg), cefoxitin (FOX, 30 µg), meropenem (MEM, 10 μg), tetracycline (TE, 30 μg), chloramphenicol (C, 30 μg), ciprofloxacin (CIP, 5 μg), nalidixic acid (NA, 30 μg), pefloxacin (PEF, 5 μg), gentamicin (CN, 10 μg), streptomycin (S, 10 μg), trimethoprim-sulfamethoxazole (SXT, 23.75/1.25 μg), sulfisoxazole (ST, 300 μg), trimethoprim (TMP, 5 μg). *Escherichia coli* ATCC 25922 was used as a control strain.

ESBL-producing *Salmonella* strains were confirmed by the double-disk synergy test performed by positioning a disk of amoxicillin+clavulanic acid 30 µg between a disk of cefotaxime 30 μg and a disk of ceftazidime 30 μg [56]. The CLSI interpretive criteria for disk diffusion susceptibility testing of *Salmonella* were used [57].

### 4.3. Bacterial DNA Extraction and Plasmid Typing

Bacterial DNA was obtained by the boiling lysis method, incubating the isolated colonies in distilled water for 10 min at 100 °C. The samples were then centrifuged at 15,000× *g* for 5 min and the supernatants were used for the following reactions. 

All strains were typed by PCR-based replicon typing (PBRT) using the PBRT kit 2.0 (Diatheva, Fano, Italy) in order to identify plasmid replicons. This PBRT assay consists of eight multiplex PCRs and allows the detection of 30 replicons of the main plasmids in *Enterobacterales*. 

All PCR reactions were carried out in accordance with the manufacturer’s instructions, including positive controls. The amplicons were detected through capillary electrophoresis on the AATI Fragment Analyzer (Agilent, Santa Clara, CA, USA) using the dsDNA 906 Reagent kit (Advanced Analytical, Ankeny, IA, USA). This amplicon analysis allows the combination of two multiplex PCRs in the same lane, resolving up to eight peaks, as previously published [58]. The positive peaks were successively analysed using the tool “PBRT plugin” [58] developed in cooperation with the Advanced Analytical Company that allows automatic peak calling and the recording of positive replicons.

### 4.4. PCR screening of Tetracycline Resistance Genes

*Salmonella* strains with tetracycline-resistant (n. 83) and intermediate (n. 3) phenotypes were investigated to detect the presence of *tet*A, *tet*B, *tet*C, *tet*D, *tet*G, and *tet*M genes by PCR. All PCR reactions were carried out following protocols indicated by the European Reference Laboratory for Antimicrobial Resistance (EURL-AR, Technical University of Denmark, National Food Institute) [59] (Table 3). 

Positive control strains for the above-mentioned *tet* genes were also provided by the EURL-AR (Table 4). 

The PCR products were separated by electrophoresis on a 1.5% (*w/v*) agarose/*midori* green advance color gel (Resnova, Rome, Italy) and finally recorded using UV transillumination. A 100 bp DNA ladder (GeneRuler 100 bp Plus, ThermoFisher Scientific, Waltham, US) was included in all agarose gels as a molecular weight standard.

### 4.5. Statistical Analysis

The association between the presence of the Inc plasmid group and the resistance phenotype was analyzed by the Hypothesis test–difference between two frequencies in large samples. A *p*-value < 0.05 was considered significant.

## 5. Conclusions

In conclusion, this study shows that a heterogeneous *Salmonella* serovars population colonizes the food chain with the ability to reach consumers and cause serious illnesses. Isolates analysed here were characterised by a high rate of resistance to a wide panel of antibiotics, and particular attention was focused on tetracyclines considering the extensive use of this agent in veterinary medicine. The presence of MDR strains carrying genetic determinants on mobile genetic elements is a matter of concern for the spread of resistance in food settings. 

In this context, the monitoring of different serovars along the food chain and a better knowledge of them from a molecular point of view are essential to managing risks associated with consumer health.

Greater coordination across all sectors of the food chain is needed to fight antibiotic resistance, and the implementation of molecular typing methods within routinary screening programmes can become an important tool of the One Health preventive approach.

## Figures and Tables

**Figure 1 antibiotics-11-00725-f001:**
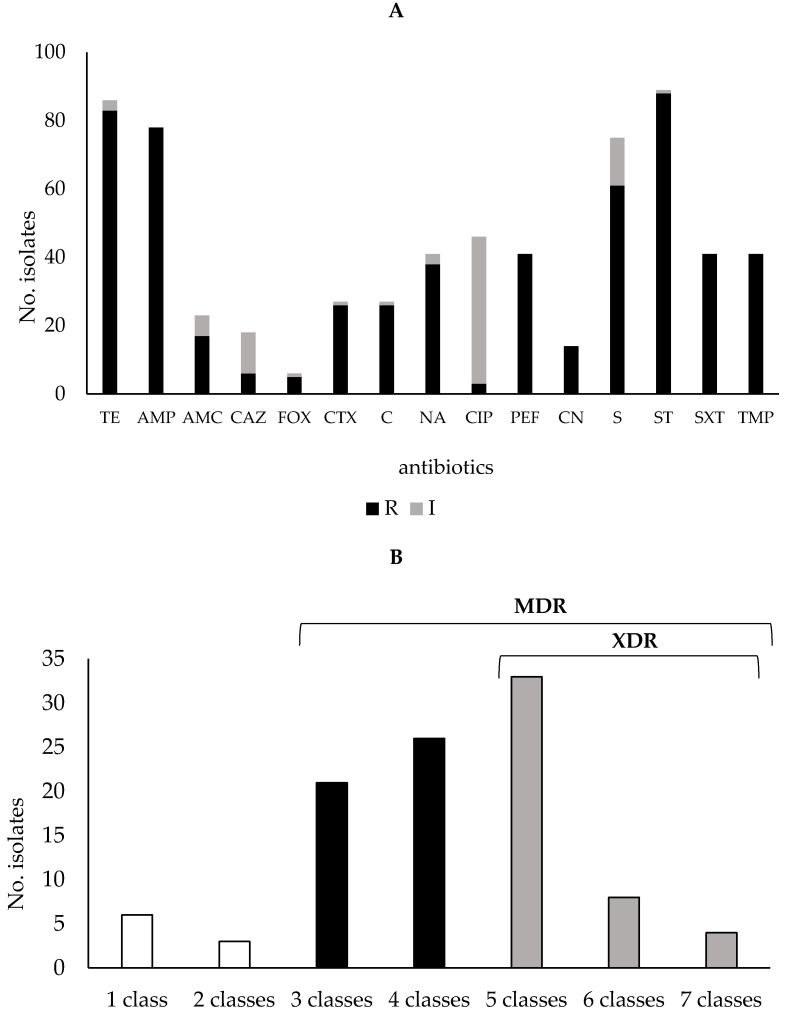
Resistance rate of *Salmonella* strains analysed in this study. (**A**) Strains were resistant to one or more antibiotic classes, and the major resistance was detected against sulfisoxazole and tetracycline. R: resistant (black), and I: intermediate (grey) resistant strains. AMP, ampicillin; CTX, cefotaxime; CAZ, ceftazidime; AMC, amoxicillin+clavulanic acid; FOX, cefoxitin; TE, tetracycline; C, chloramphenicol; CIP, ciprofloxacin; NA, nalidixic acid; PEF, pefloxacin; CN, gentamicin; S, streptomycin; SXT, trimethoprim-sulfamethoxazole; ST, sulfisoxazole; TMP, trimethoprim. (**B**) Distribution of Multi-and Extensively Drug-Resistant *Salmonella* strains. The major part of strains were resistant to three or more antibiotic classes (multidrug-resistant; MDR) whereas 44% were resistant to at least one agent in all but two or fewer antimicrobial categories (extensively drug-resistant; XDR) (**B**).

**Figure 2 antibiotics-11-00725-f002:**
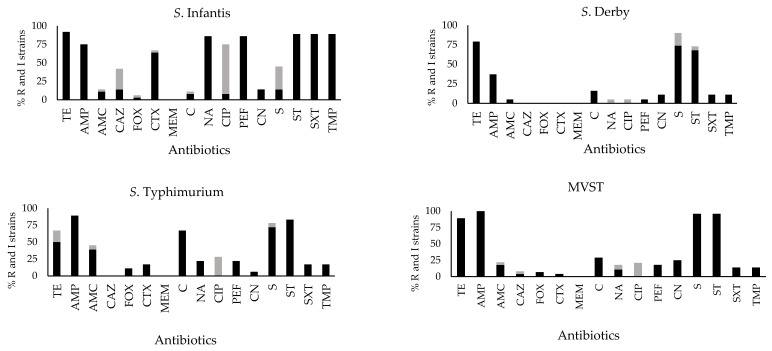
AMR profile of *Salmonella* strains investigated in this study, according to serovar. R: resistant (black), and I: intermediate (grey) strains. MVST: monophasic variant of *S*. Typhimurium. AMP, ampicillin; CTX, cefotaxime; CAZ, ceftazidime; AMC, amoxicillin+clavulanic acid; FOX, cefoxitin; MEM, meropenem; TE, tetracycline; C, chloramphenicol; CIP, ciprofloxacin; NA, nalidixic acid; PEF, pefloxacin; CN, gentamicin; S, streptomycin; SXT, trimethoprim-sulfamethoxazole; ST, sulfisoxazole; TMP, trimethoprim.

**Figure 3 antibiotics-11-00725-f003:**
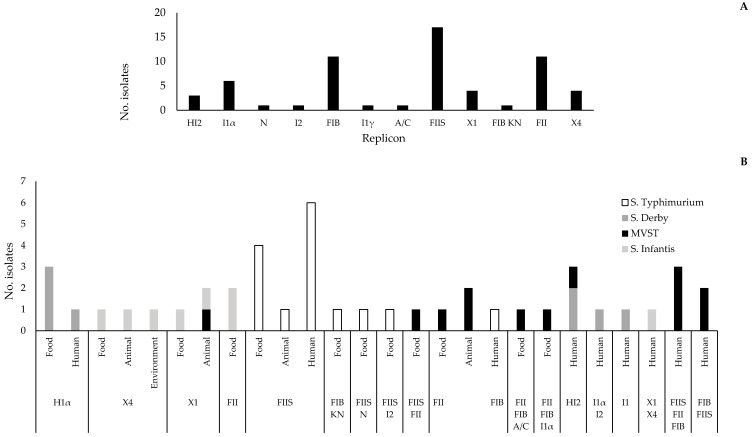
Replicons detected by PBRT and distribution of replicon patterns among *Salmonella* serovars isolated from different sources. (**A**) Replicons detected by PBRT. (**B**) Replicon patterns and distribution among the serovars of *Salmonella* strains isolated from different sources. MVST: monophasic variant of *S*. Typhimurium.

**Table 1 antibiotics-11-00725-t001:** Source and serotypes of *Salmonella* strains.

Source	Serovars	Strain n.
Animals	*S.* Infantis	8
	*S.* Derby	1
	*S.* Typhimurium	1
	MVST	6
Environment	*S.* Infantis	5
	*S.* Derby	1
	*S.* Typhimurium	0
	MVST	3
Foods	*S.* Infantis	11
	*S.* Derby	8
	*S.* Typhimurium	7
	MVST	9
Humans	*S.* Infantis	12
	*S.* Derby	10
	*S.* Typhimurium	10
	MVST	10
Total		102

**Table 2 antibiotics-11-00725-t002:** *tet* genes detected in tetracycline-resistant and intermediate strains, and serovars.

*tet* Genes	Resistant Strains	Intermediate Strains
*tet*A	10 2 *S*. Derby, 7 *S*. Infantis, 1 *S*. Typhimurium	0
*tet*B	27 1 *S*. Derby, 24 MVST, 2 *S*. Typhimurium	0
*tet*C	0	0
*tet*D	0	0
*tet*G	3 *S*. Typhimurium	2 *S*. Typhimurium
*tet*M	1 MVST	0

**Table 3 antibiotics-11-00725-t003:** Primer and amplicon features of PCR assays used for *tet* gene characterization.

Target Gene	Primer Sequence	Amplicon Size (bp)	Tm (°C)	Ref.
*tetA*	5′-GTAATTCTGAGCACTGTCGC-3′	956	57	[60]
	5′-CTGCCTGGACAACATTGCTT-3′			
*tetB*	5′-CTCAGTATTCCAAGCCTTTG-3′	414	52	[60]
	5′-ACTCCCCTGAGCTTGAGGGG-3′			
*tetC*	5′-GGTTGAAGGCTCTCAAGGGC-3′	505	65	[60]
	5′-CCTCTTGCGGGATATCGTCC-3′			
*tetD*	5′-CATCCATCCGGAAGTGATAGC-3′	436	57	[61]
	5′-GGATATCTCACCGCATCTGC-3′			
*tetG*	5′-GCAGCGAAAGCGTATTTGCG-3′	662	62	[62]
	5′-TCCGAAAGCTGTCCAAGCAT-3′			
*tetM*	5′-GTTAAATAGTGTTCTTGGAG-3′	657	45	[62]
	5′-CTAAGATATGGCTCTAACAA-3′			

**Table 4 antibiotics-11-00725-t004:** Positive control strains used for *tet* gene PCR-based characterisation provided by the European Reference Laboratory for Antimicrobial Resistance (EURL-AR, Technical University of Denmark, National Food Institute).

Species	Strain Name	*tet* Gene
*E. coli*	*tet*A, NCTC 50078	*tet*A
*E. coli*	*tet*B, CSH50::Tn10	*tet*B
*E. coli*	*tet*C DO7 pBR 322, Tet	*tet*C
*E. coli*	*tet*D C600 psl 106, Tet *tet*Dx2	*tet*D
*S.* Typhimurium	P502212 DT104 sul1 and *tet*G	*tet*G
*Staph. intermedius*	2567, chromosomal *tet*M	*tet*M

## Data Availability

The data presented in this study are present within the article.

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
