# Peer review of "Prevalence, Antibiotic-Resistance, and Replicon-Typing of Salmonella Strains among Serovars Mainly Isolated from Food Chain in Marche Region, Italy"

_antibiotics, 2022, doi:10.3390/antibiotics11060725_

Round 1

Reviewer 1 Report

Congratulations for your work. The study is well designed, methods employed well described, and most importantly, results are of wide relevance, even more considering that they lay the ground for further deepening on the implications of those genes and plasmids in antibiotic resistance not only of Salmonella spp. but also other species of relevance in the field of foodborne pathogens. Nevertheless, here you have some little issues that might help you to even improve your paper.

Line 62. Sites? Maybe órganos?

Lines 75-80. Maybe, it worths mentioning that the mutations have also vertical transference.

Section 2.2. Which were the values you stamdarize for the classification of Salmonella serovars as intermediate resistant or resistant to antibiotics?

Lines 136 and 137. Please, change “resistant” for “resistance”.

Line 145. Intermediate resistant strains.

Line 168. Were negative because they were negative…

Lines 265 and onwards. I miss some info about the consumption of antibiotics in Veterinary Medicine in Europe, just to compare the match between the most detected resistances and the most used antibiotics.

Line 266. There is a point lack after the citation number 35.

Line 267. To keep the associated risk

Line 268: There is a point lack after de citation number 36.

Lines 265-270. Please, re-order the paragraph and consider its merging with the next one. Re-organization might be done moving the las sentence after the first one.

Line 272: in accordance with the last…

Line 275. As a suggestion, corroborating could be changed for a synonym such as “matching”.

Line 278. Please, change “can” by “could”

Line 279. Please, change the comma after the word “pumps” for “or”

Line 286. Please, kindly change “First” for “Firstly”, as in the next sentence you use the nexus “Secondly”

Line 326. Kindly change “need to monitor” for “need of monitoring”.

Line 331. Please, consider the change of “to understand as resistant bacteria” for “to understand how resistant bacteria”.

Lines 351-352. Where did you collected these samples from? It is clear that they are collected from animals, but it is not clearly explained whether they are collected from the hair, feathers, rectum, faeces, etc. Regarding sample collection, I also miss the number of samples collected, and in case they pertain to a bigger number as they were side-samples collected for official control, why were those 102 selected. Moreover, it would be good to know whether resistant strains were coming from the same or different niches.

Line 370. There is one space lack after “1.25” and before “μg”.

Line 396. Please, note that in some sections you refer Salmonella as “Salmonella spp. strains” (for instance, in line 396) and in some others as “Salmonella strains” (for instances, in lines 363 and 372)

Reviewer 2 Report

The manuscript provides information on the investigation of antibiotic-resistance of Salmonella Serovars isolated from different sources, including humans, in Italian Region. Plasmid profiles of investigated bacteria are also given. The study aims to provide better insight in the prevalence of resistant Salmonella and their spread in foods and other sources which can be important for risk assessment and management concerning human health.

The Introduction and discussion are well written but some issues regarding the results should be addressed.

In results section 2.2. should be revised. Figure captions for figure 1 do not correspond with the figure itself. Description for A and B part is mixed. Also, B part (in figure) is not well explained.

In the text, lines 135-137 it is not clear what is meant by XDR, what is then meant by intermediate resistance. From line 139-140 it could be concluded that there is only one intermediate strain.

Lines 144-145 the last part of the sentence is not clear. Again, it is not clear what is meant by Intermediate stains. How many are there?

In Figure 3. Letters A and B should be written for each graph. Figure captions should be revised as well, the abbreviations for antibiotics are unnecessary since they are not mentioned in the figure itself.

I suggest mentioning Supplementary Figure in the text where appropriate.

Line 121. Full information for abbreviation ESBL should be given (it is given at the and in materials and methods section) but it is mentioned here first.

Round 2

Reviewer 2 Report

The authors have addressed all the issues and the manuscript can now be accepted for publication.